# GPT4Tools: Teaching Large Language Model to Use Tools via Self-instruction

**Rui Yang**[1][*][‡]   **Lin Song**[2][*][†]   **Yanwei Li**[3]   **Sijie Zhao**[2]   **Yixiao Ge**[2]   **Xiu Li**[1]   **Ying Shan**[2]

[1]Tsinghua Shenzhen International Graduate School, Tsinghua University
[2]Tencent AI Lab   [3]Chinese University of Hong Kong
rayyang0116@gmail.com   ronnysong@tencent.com

*The essential difference between humans and animals is that humans are capable of making and using tools.*

—Friedrich Engels

## Abstract

This paper aims to efficiently enable Large Language Models (LLMs) to use multi-modal tools. Advanced proprietary LLMs, such as ChatGPT and GPT-4, have shown great potential for tool usage through sophisticated prompt engineering. Nevertheless, these models typically rely on prohibitive computational costs and publicly inaccessible data. To address these challenges, we propose the GPT4Tools based on self-instruct to enable open-source LLMs, such as LLaMA and OPT, to use tools. We generate an instruction-following dataset by prompting an advanced teacher with various multi-modal contexts. By using the Low-Rank Adaptation (LoRA) optimization, our approach facilitates the open-source LLMs to solve a range of visual problems, including visual comprehension and image generation. Moreover, we provide a benchmark to evaluate the ability of LLMs to use tools, which is performed in both zero-shot and fine-tuning ways. Extensive experiments demonstrate the effectiveness of our method on various language models, which not only significantly improves the accuracy of invoking seen tools but also enables the zero-shot capacity for unseen tools. The code and demo have been available at https://github.com/AILab-CVC/GPT4Tools.

## 1   Introduction

Recent advances in large language models (LLMs), such as GPT-3 [1], InstructGPT [2], and GPT-3.5 [3], have demonstrated substantial potential in the area of zero-shot learning and logical reasoning. These models are typically trained on a large volume of text-only data, primarily sourced from the internet. However, as promising as they may seem, these advanced proprietary LLMs [3, 4] have significant limitations. One of the major hindrances is the high computational cost associated with these models, which may not be affordable or accessible to many scenarios. Additionally, these models typically depend on specialized data, such as source code and conversation history, which are not easily available to the public.

Instead of solely focusing on language processing, many recent researches [5, 6] attempt to bridge the gap between language models and multi-modal models. Intelligent agents like Visual ChatGPT [5] and MMREACT [6] have made efforts to meet this goal by sophisticated prompt engineering. These agents utilize a pre-defined template to create instructions that vision-language foundation models can execute. Although these approaches have led to impressive results, the primary process of instruction

---

[*]Equal contribution. ‡ Work done during an internship at Tencent.
[†]Corresponding author.

37th Conference on Neural Information Processing Systems (NeurIPS 2023).

Table 1: Comparison of related works. 'LM' is the language model. 'Mechanism' denotes how the language model learns to invoke tools. 'Unseen' indicates the zero-shot capability on unseen tools.

| Method | LM | Mechanism | Teacher | Multi-Modal | Unseen |
|---|---|---|---|---|---|
| Lazaridou et al. [10] | Gopher-280B [15] | prompt | ✗ | ✗ | ✗ |
| ToolFormer [11] | GPT–J (6B) [16] | self-instruct | ✗ | ✗ | ✗ |
| Visual ChatGPT [5] | GPT-3.5 (175B) [3] | prompt | ✗ | ✔ | ✔ |
| MMREACT [6] | GPT-3.5 (175B) [3] | prompt | ✗ | ✔ | ✔ |
| GPT4Tools (**ours**) | Vicuna-13B [12] | self-instruct | ✔ | ✔ | ✔ |

decomposition is heavily based on GPT-3.5 [3], which is expensive and not publicly available, thus limiting further advancements. In addition, equipping these agents with the capability to use tools requires a large amount of data [3]. This brings up an open question: *how to efficiently enable a primitive language model to use multi-modal tools?*

To achieve it, different from previous studies [7–11], we explore a new perceptive as illustrated in Table 1. We propose a simple yet effective method, called GPT4Tools, designed to empower open-source LLMs with the ability to use tools via self-instruct from advanced LLMs. To be specific, we construct an instruction dataset by prompting advanced teachers, such as GPT-3.5 [3], conditional on visual contents and tool descriptions, which results in a great deal of tool-related instructions. Unlike Toolformer [11], our method can utilize visual content description to improve data diversity significantly. Furthermore, with the generated instruction-following dataset, we employ Low-Rank Adaptation (LoRA) to fine-tune the primitive language models including Vicuna [12], LLaMa [13], and OPT [14]. Besides intrinsic language abilities, GPT4Tools-based language models are also able to solve a variety of visual problems by using tools. The tasks involve visual comprehension and image generation, such as object grounding and segmentation, generating and instructing images, and visual question answering (VQA). The proposed GPT4Tools not only significantly improve the accuracy of LLMs to invoke seen tools but also enable the zero-shot capacity for unseen tools in a zero-shot manner.

Furthermore, we propose an evaluation metric to assess the effectiveness of LLMs in utilizing tools across diverse tasks. With this metric, two human-curated validation sets are constructed to evaluate the LLMs in zero-shot and fine-tuning ways, offering a comprehensive measure of the ability to use tools. To demonstrate the effectiveness of GPT4Tools, we conduct extensive experiments on various language models. The results show the efficacy of teaching LLMs when and how to use tools. Specifically, the Vicuna-13B fine-tuned on our GPT4Tools achieves 9.3% absolute gains in successful rate over GPT-3.5 [3] that acquires tool priors in context. In addition, the fine-tuned Vicuna-13B shows a solid capacity to invoke unseen tools, wihich can be comparable to GPT-3.5's success rate.

Our GPT4Tools stands distinct from previous and concurrent studies [5–11] in three ways. First, our method enables primitive open-source language models to use tools, eliminating the dependence on advanced proprietary LLMs like ChatGPT. Second, we design a new approach based on multi-modal contexts for self-instruction and augmentation, which significantly promote multi-modal tool usage and can be deployed in different directions. Third, we propose a new benchmark to assess the effectiveness of using tools, and our method shows remarkable improvements.

## 2 Related Work

**Vision and Language Model.** In the quest to achieve multi-modal models capable of addressing both language and vision tasks, several studies [17–22] have explored methods to enable language models to comprehend visual input. These include techniques such as transforming images into discrete textual representations [17, 18] or projecting continuous image features into the textual feature space [23–26]. Concurrently, other research has been dedicated to the development of generalist models [19, 20, 22, 21], which permit a model to simultaneously input images and text, eliminating the necessity for a projection process. For instance, OFA [27] devised a unified sequence-to-sequence decoding architecture applicable to language and object detection tasks. Similarly, Pixel2Pixel [21] converted the outcome of visual comprehension tasks into a series of discrete tokens akin to language tasks. Gato [22] brought together a range of vision and control tasks into a sequential prediction issue,

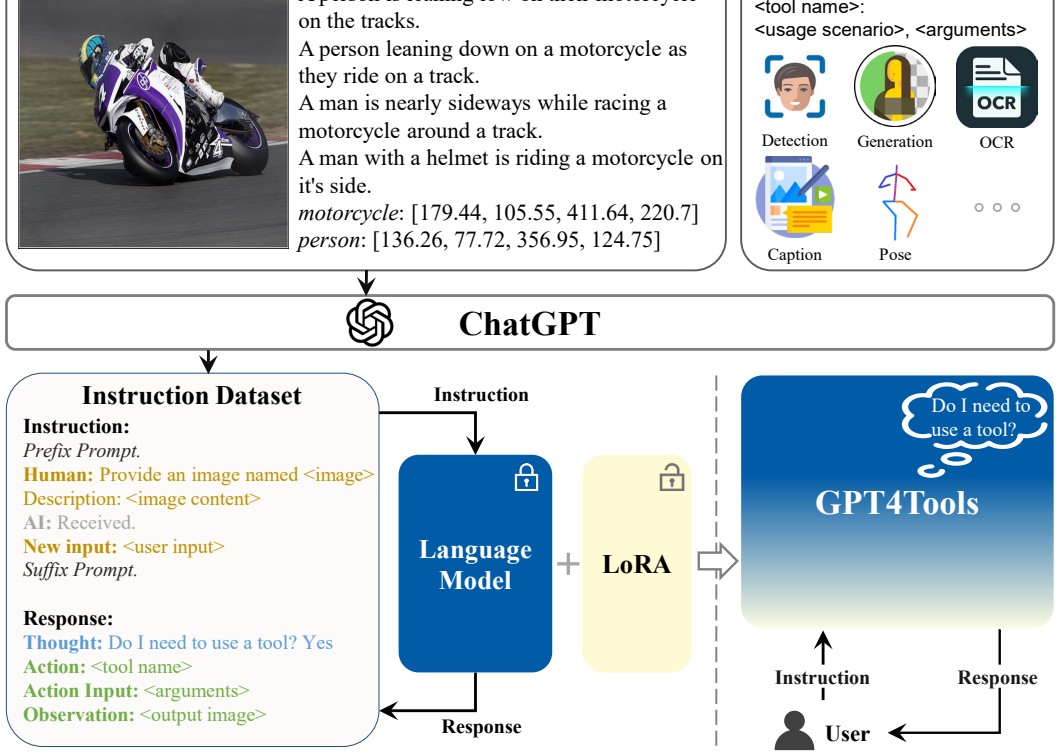

Figure 1: Diagram of the GPT4Tools. We prompt the ChatGPT with image content and definition of tools in order to obtain a tool-related instruction dataset. Subsequently, we employ LoRA [38] to train an open-source LLM on the collected instruction dataset, thus adapting the LLM to use tools.

while UViM [28] and Unified-IO [20] advocated for the learned discrete codes as a means to unify an array of vision tasks. By contrast, we equip the language model with diverse specialized multi-modal tools to process distinct vision tasks. This approach not only promotes the scalability of the model for various tasks but also avoids the issue of forgetfulness stemming from repeated fine-tuning.

**Instruction Tuning.** Recent studies [29, 2, 30–33] have turned out that pre-trained language models could follow natural language instructions and complete various real-world tasks if they are tuned on specific instruction-following data. Notably, InstructGPT [2], FLAN-T5 [31], OPT-IML [33] demonstrated remarkable performance on specific tasks after being fine-tuned with instruction data. In order to release the cost of human-written instructions, Self-Instruction [34] found that the instruction-following capabilities of language models can be enhanced by turning on their own generated instruction data. More importantly, this approach inspired a feasible means to improve the zero- and few-shot abilities of language models, i.e., distilling off-the-shelf language models using instructional data from strong GPT-3.5 [3] or GPT-4 [4]. As a result, many recent works [12, 35–37] tried to construct excellent language models for various applications based on the LLaMA [13]. For instance, Stanford-Alpaca has employed 52K instructions generated by GPT-3.5 [3] to construct an exceptional dialogue model. LLaVa [37] has adopted GPT-3.5 [3] and GPT-4 [4] to incorporate instruction-following data related to visual content. In this paper, we use GPT-3.5 [3] to construct tool-related instruction datasets, thereby allowing other language models to acquire tool usage capabilities.

**Tool Usage.** In the Natural Language Processing (NLP) community, several arts [7–11] sought to endow language models with the ability to use tools. For instance, Komeili et al. [7] proposed to generate conversation responses conditioned on the results of the search engine. LaMDA [9] created a set of tools (comprising an information retrieval system, a calculator, and a translator) to avoid plausible outputs. Lazaridou et al. [10] utilized few-shot prompting on Gopher-280B [15] to enable the search engine to ground its output in factual and current information. Similarly, Visual ChatGPT [5] and MMREACT [6] prompted ChatGPT to invoke visual foundation models. In addition,

ToolFormer [11] used self-instruction and bootstrapping to teach GPT-J (6B) [16] using five tools, which include a question and answer system, a calculator, a search engine, a machine translation system, and a calendar. On the contrary, we focus on using the GPT-3.5 model as a powerful teacher to distill off-the-shelf language models and enable them to access many visual models.

## 3 Method

Large language models (LLMs) [1, 14, 15] have shown remarkable in-context learning abilities. Among them, GPT-3.5 [3] and GPT-4 [4] are proven to effectively perform text-annotation tasks [39] or instruct other models to follow instructions of specific domains [40, 35, 12, 37]. Inspired by these findings, we propose to enable off-the-shelf language models to acquire tool usage capabilities by taking GPT-3.5 as a powerful teacher. Specifically, we utilize GPT-3.5 to generate tools-related instruction-following data, which is then used to tune the language model. This process offers language models the ability to access multi-modal information by invoking visual models. Furthermore, we propose an evaluation metric to assess the tool-use ability of the given language model. In the following, we elaborate on the data generation, instruction tuning, and evaluation metric in turn.

### 3.1 Dataset Construction

**Data Generation.** Figure 1 illustrates the process of generating tool-related instruction dataset. Given an image, we construct the image content $X_C$ according to the captions and bounding boxes, which is a straightforward means of establishing connections between an image and a language model [17, 18]. Conditioned upon the $X_C$, we provide the GPT-3.5 [3] ($M_T$) with a tool-related prompt $P_t$ whereby attaining a large number of instruction-following data:

$$Y \sim M_T(P_t|X_C). \tag{1}$$

The $P_t$ comprises the system message, the definition of tools (`<tool name>`: `<usage scenario>`, `<arguments>`), and the suffix prompt which encourages $M_T$ to generate visual instructions and desired outputs. $Y$, the outcome of $M_T$, consists of $N$ instruction-output pairs $\{y^1, y^2, ..., y^N\}$, where $y_i$ has the format of "`<instruction>`, `<tool name>`, `<arguments>`", and $N$ is the number of defined tools. As each input of $M_T$ is grounded to the image content $X_C$, the generated instructions are inherently connected to the image, thus avoiding arbitrary generation. In detail, the $X_C$ consists of ground-truth captions and bounding boxes with tags corresponding to the images. The rich variability of the image brings up a higher diversity of instructions when compared to imagined ones. To provide contrast, we also collect instruction follow-up data without image content $X_C$, which is similar to ToolFormer [11]. As depicted in Figure 2, without image context priors, GPT-3.5 tends to generate objects of visual instructions towards a small subset, which is reflected in t-SNE as sparser clusters. On the contrary, instructions generated with image-conditioned prompts are notably informative and diverse due to changes in the image content, reflected in the visualization as denser and more widely distributed results. The language model tuned by image-conditioned data is more robust than models without the image content (Table 3).

**Data Formation.** Upon the collected raw dataset (70K items), we apply a filtering process to remove duplicate instructions, incorrectly formatted instructions, calls with incorrect tool names, and calls with incorrect tool-arguments formats. This step results in 41K retained items. Subsequently, we transform the retained data into an instruction-response format utilizing a standardized template as shown in the bottom-left corner of Figure 1. This procedure produces a new dataset, denoted as $Y_S^+$. The instruction component of $Y_S^+$ incorporates a *prefix prompt* that encompasses system messages and tool definitions,

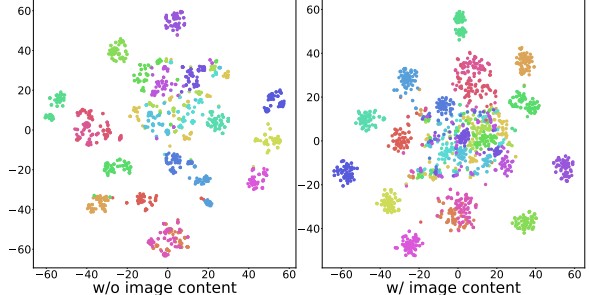

Figure 2: t-SNE[1] visualization for instruction data with and without image content.

---
[1]The visual instruction and tool arguments are embedded by Sentence-BERT [41]. We set the temperature for generation without image content as 0.9 to avoid constant outcomes.

| (a) Single-turn sample | (b) Negative sample | (c) Context sample |

**Instruction:**
*Prefix Prompt.*
**Human:** Provide an image named *example.png*. Description: a huge crowd of people at a sky resort Understand the image using tools.
**AI:** Received.
**New input:** Generate a real image of a ski resort with people and winter clothing from segmentations and text description.
*Suffix Prompt.*

**Response:**
**Thought:** Do I need to use a tool? Yes
**Action:** Segment the Image
**Action Input:** *example.png*
**Observation:** *output_1.png*
**Thought:** Do I need to use a tool? Yes
**Action:** Generate Image Condition On Segmentations
**Action Input:** *output_1.png*, A ski resort with people and winter clothing
**Observation:** *output_2.png*
**Thought:** Do I need to use a tool? No
**AI:** Result saved as output_2.png

**Instruction:**
*Prefix Prompt.*
**Human:** Provide an image named *example.png*. Description: The motorcy- clists are travelling down the street together. Understand the image using tools.
**AI:** Received.
**New input:** Select the best definition of "attribute".
*Suffix Prompt.*

**Response:**
**Thought:** Do I need to use a tool? No
**AI:** An attribute can be defined as a quality, trait, or characteristic inherent to a person, thing, or object. In the context of database management, it refers to a specific value or characteristic of a variable or column within a record. In programming or web development, an attribute defines properties, settings or characteristics of an element in a markup language or code.

**Instruction:**
*Prefix prompt.*
**Human:** Provide an image named *example.png*. Description: many people are at their laptops working together Understand the image using tools.
**AI:** Received.
**New input:** Generate a new real image of a conference room full of people working on their laptop computers from this depth map.
*Suffix prompt*
**Thought:** Do I need to use a tool? Yes
**Action:** Predict Depth On Image
**Action Input:** *example.png*
**Observation:** output_1.png

**Response:**
**Thought:** Do I need to use a tool? Yes
**Action:** Generate Image Condition On Depth
**Action Input:** *output_1.png*, The image depicts a conference room full of people working on their laptop computers.
**Observation:** *output_2.png*
**Thought:** Do I need to use a tool? No
**AI:** Result saved as output_2.png

Figure 3: Samples of the single-turn instruction, negative instruction, and contextual instruction.

`<image content>` that denotes the image content, `<user input>` that is replaced with the generated visual instruction, and a *suffix prompt* designed to prompt the language model to reply the user input using given tools. The response in $Y_S^+$ comprises 4 elements: (1) *Thought*, meaning the model's cognition when to use tools; (2) *Action*, signifying which tools the model will use or action the model will take; (3) *Action Input*, representing arguments of the selected tool; and (4) *Observation*, reflecting outcomes of the used tool. A sample from $Y_S^+$ is presented in the Figure 3 (a).

**Data Augmentation.** Although we have successfully acquired instruction-following data related to the tool usage, this simplistic format lacks complexity and depth in both instructions and responses. To tackle this challenge, we augment the generated data from two perspectives:

- *Negative samples.* The generated instructions primarily focus on tool usage, i.e., the decision after the *Thought* is always "Yes". Consequently, there is a potential risk that the fine-tuned model overfits such a decision. When the user instruction is not associated with the tool usage, the fine-tuned model may erroneously execute irrelevant actions by invoking unnecessary tools. To mitigate this issue, we synthesize negative samples $Y_S^-$ by selecting conversation data from the existing dataset [40] and converting them into the required template, as illustrated in Figure 3 (b). By tuning with $Y_S^+ \cup Y_S^-$, the model can accurately decide when to use tools.

- *Context samples.* The generated instructions adopt a standard and fixed single-tune format, which lacks a contextual structure. Thus, as shown in Figure 3 (c), we augment the dataset by cutting off the chain of action. We also randomly select multiple instructions from $Y_S^+ \cup Y_S^-$ and reformat them into multi-turn conversation data. In this way, we synthesize the contextual instruction-following data $Y_S^c$, enabling the tuned model to call tools within the given context.

So far, we have constructed the tool-related instructional dataset, including positive samples, negative samples, and context samples: $Y_S = Y_S^+ \cup Y_S^- \cup Y_S^c$.

## 3.2 Instruction Tuning

Based on the dataset $Y_S$, we tune the off-the-self language model using its original auto-regressive training objective. To make the tuning feasible, we leverage LoRA [38] optimization, which freezes the language model and only optimizes rank decomposition components of the Transformer layers.

For a sequence with $L$ tokens, we compute the probability of the target response $X_r$ by:

$$p(X_r|X_C, X_{inst}) = \prod_{i=1}^{L} p_\theta(x_i|X_C, X_{inst}, x_{1:i-1}), \tag{2}$$

where $X_{inst}$ denotes the instruction tokens; and $\theta$ is the trainable parameters. In practice, *prefix prompt* and *suffix prompt* are also involved but we here skip them for better readability.

### 3.3 Evaluation Approach

Numerous benchmarks [42, 43, 8, 44] typically utilize human-annotated datasets to evaluate the performance of a model. For the purpose of measuring the tool-usage capacity of the language model, we construct an evaluation dataset following the same procedures detailed in § 3.1 and manually verify the accuracy of each item. This evaluation dataset is partitioned into two components: the first part (validation set) has the same ingredients as the training set, encompassing 23 tools; the second part (test set) comprises 8 novel tools absent from the training set. We will use the validation set to validate whether the model can adhere to user commands correctly after tuning with the training set. The test set will verify whether the model can generalize to new tools after tuning. Based on the human-annotated evaluation dataset with $N$ instructions, we design a successful rate to measure the model's performance from three aspects:

- **Successful Rate of Thought** ($\mathrm{SR}_t$) measures whether the predicted decision matches the ground-truth decision. It is calculated as $\mathrm{SR}_t = \frac{1}{N}\sum_{i=1}^{N} \mathbb{I}(\tau_i)$, where $\tau_i$ signifies a singular process. If the thought is correct, $\mathbb{I}(\tau_i)$ is equal to 1, and 0 otherwise.

- **Successful Rate of Action** ($\mathrm{SR}_{act}$) measures whether the predicted tool name is in agreement with the name of the ground truth tool. It is calculated as $\mathrm{SR}_{act} = \frac{1}{N}\sum_{i=1}^{N} \mathbb{I}(\alpha_i)$, where $\alpha_i$ denotes the matching process for the tool names. In cases where the predicted tool name matches the pre-defined name, $\mathbb{I}(\alpha_i)$ equals 1, and 0 otherwise.

- **Successful Rate of Arguments** ($\mathrm{SR}_{args}$) evaluates whether the predicted arguments match the ground-truth arguments. It can be calculated using the following equation:

$$\mathrm{SR}_{args} = \frac{1}{N}\sum_{i=1}^{N}\eta_i, \text{ where } \eta_i = \frac{1}{K}\sum_{j}^{K}\eta_{i,j}. \tag{3}$$

Here, $\eta_i$ denotes a sequence of arguments encompassing both the image path and the input text. For instance, ControlNet [45] needs the image path saved conditions (e.g., the pose map, depth map, or segment map) and the input text described user commands. $K$ represents the quantity of arguments in $\eta_i$. When the argument belongs to the image path, $\eta_{i,j}$ equals 1 if the predicted and ground-truth image paths share the same suffix, and 0 otherwise. When the argument is the input text, $\eta_{i,j}$ is equal to the BLEU score between the predicted and the ground truth text.

- **Successful Rate** (SR) measures whether a chain of actions are executed successfully, which requires the correctness of thought, tool name, and tool arguments at the same time:

$$\mathrm{SR} = \frac{1}{N}\sum_{i=1}^{N} \mathbb{I}(\tau_i) \cdot \mathbb{I}(\alpha_i) \cdot \mathbb{I}(\eta_i > 0.5) \tag{4}$$

Additionally, when a procedure comprises two consecutive actions, the SR equals $100\%$ only if both actions are executed correctly.

## 4 Experiments

### 4.1 Implementation Details

We employ the ChatGPT (`gpt-3.5-turbo`) [3] as the teacher model to generate the raw instruction-following data. Since this study focused on teaching the off-the-self language models to use tools instead of prompt engineering, we adopted a methodology outlined in the Visual ChatGPT [5] to construct tool-related prompts. Our tool pocket consists of 31 tools, including the 23 tools defined

Table 2: Comparison of different language models. The zero-shot prediction is adopted for unseen tools and the models without GPT4Tools.

| Model | GPT4Tools | Validation (seen tools) | | | | Test (unseen tools) | | | |
|---|---|---|---|---|---|---|---|---|---|
| | | $SR_t$ | $SR_{act}$ | $SR_{args}$ | SR | $SR_t$ | $SR_{act}$ | $SR_{args}$ | SR |
| GPT-3.5 [3] (text-davinci-003) | ✗ | 93.5 | 96.1 | 78.0 | 84.8 | 99.5 | 99.5 | 91.5 | 91.5 |
| OPT-13B [14] | ✗ | 1.1 | 1.2 | 0.0 | 0.0 | 0.0 | 0.0 | 0.0 | 0.0 |
| | ✔ | **99.4** | **98.3** | **89.2** | **93.2** | **97.8** | **89.6** | **84.0** | **78.6** |
| LLaMa-13B [13] | ✗ | 20.4 | 15.7 | 16.5 | 3.2 | 16.1 | 17.6 | 21.7 | 2.0 |
| | ✔ | **77.3** | **74.9** | **71.4** | **66.4** | **74.2** | **72.2** | **70.9** | **69.9** |
| Vicuna-13B [12] | ✗ | 69.2 | 25.1 | 25.2 | 12.4 | 84.4 | 43.7 | 46.7 | 26.2 |
| | ✔ | **98.7** | **97.6** | **91.4** | **94.1** | **98.2** | **97.0** | **92.2** | **90.6** |

in Visual ChatGPT [5] and 8 extra tools (please refer to Appendix for detailed tool names). During generation, all image information utilized in GPT4Tools is sourced from the training set of COCO [43]. After generation, the training set comprises $71K$ instruction-response pairs, wherein all instructional data is related to the 23 tools. We divided the human-annotated evaluation dataset into two parts: the validation set and the test set. The validation set contains the same tools as the training set, with approximately 50 items associated with each tool. The test set includes tools that are not present in the training set (further details provided in Appendix).

Based on the collected data, we tuned language models (LLaMA [13], Vicuna [12], and OPT [14]) with LoRA [38] technology. Specifically, we equipped the projection layers of query, key, value, and output with LoRA layers. The LoRA attention dimension and scaling alpha were set to 16. While the language model was kept frozen, the LoRA layers were optimized using the AdamW [46]. All models were fine-tuned over 3 epochs, with a batch size 512. The learning rate was set to $3 \times 10^{-4}$, and the maximum length of new tokens was restricted to 2048. Unless otherwise specified, we used Vicuna-13B for the ablation experiments.

## 4.2 Main Result

**Can instruction datasets teach language model using tools?** The outcomes of GPT-3.5 [3], OPT-13B [14], LLaMA-13B [13], and Vicuna-13B [12] are presented in Table 2. GPT-3.5 is considered analogous to Visual ChatGPT [5]. Upon prompting GPT-3.5 with tool-associated instructions, it can attain a SR of $84.8\%$ on the validation set, thereby underscoring its zero-shot ability to follow a standardized format and utilize tools effectively. Notably, OPT-13B fails to invoke tools with the prompts alone. In contrast, LLaMA-13B and Vicuna-13B exhibit a certain level of comprehension of tool usage, while they still face challenges in executing a chain of actions. Specifically, LLaMA-13B achieves $3.2\%$ SR, which is absolutely lower than $SR_t$, $SR_{act}$, and $SR_{args}$. In the case of Vicuna-13B, its SR is $56.8\%$ less than $SR_t$, implying that under a zero-shot setup, Vicuna-13B displays commendable discernment in determining when to use tools within a given context. After fine-tuned with GPT4Tools, there are substantial alterations in the tool invocation competencies of each model. Specifically, the SR of OPT-13B witnessed a sharp increase from 0 to $93.2\%$. Similarly, the SR for LLaMA-13B escalates from $3.2\%$ to $66.4\%$, and Vicuna-13B's SR rises from $12.4\%$ to $94.1\%$. These outcomes unequivocally validate that the GPT4Tools developed in this study are indeed effective in instructing language models to use tools.

**Can the model be generalized to unseen tools after fine-tuning?** The right side of Table 2 shows the results when prompting a novel tool and corresponding utilization. On the test set, GPT-3.5 attaines $91.5\%$ SR in a zero-shot manner. The outcomes for other models, which are not fine-tuned on the GPT4Tools and directly invoke tools utilizing prompts, are analogous to those on the validation set. In contrast, models that are fine-tuned on the GPT4Tools dataset exhibit a degree of competence in invoking tools that have not been previously encountered (did not appear in the training set). More specifically, the fine-tuned LLaMA-13B model achieves a superior SR on new tools by a margin of $67.9\%$ when compared to the original model. The fine-tuned Vicuna-13B model demonstrates 90.6% SR on new tools, which is comparable to GPT-3.5. This observation indicates that the language model can invoke unseen tools after fine-tuned with GPT4Tools.

Table 3: Ablation study for data augmentations on the validation set. 'IC', 'CS', 'NS' denotes image content, context samples, and negative samples, respectively.

| IC | CS | NS | $SR_t$ | $SR_{act}$ | $SR_{args}$ | SR |
|----|----|----|--------|-----------|------------|-----|
|    |    |    | 70.0 | 55.7 | 51.7 | 36.9 |
| ✔  |    |    | 89.6 | 89.9 | 84.5 | 81.6 |
| ✔  | ✔  |    | 97.4 | 95.7 | 88.5 | 91.6 |
| ✔  | ✔  | ✔  | **98.7** | **97.6** | **91.4** | **94.1** |

Table 4: Ablation study for different model scales on the validation set. 7B and 13B refer to Vicuna-7B [12] and Vicuna-13B [12] models, respectively.

| Model | GPT4Tools | $SR_t$ | $SR_{act}$ | $SR_{args}$ | SR |
|-------|-----------|--------|-----------|------------|-----|
| 7B    | ✗ | 27.7 | 15.8 | 11.5 | 4.5 |
|       | ✔ | **96.2** | **94.5** | **89.8** | **92.9** |
| 13B   | ✗ | 69.2 | 25.1 | 25.2 | 12.4 |
|       | ✔ | **98.7** | **97.6** | **91.4** | **94.1** |

## 4.3 Ablation Study

**Data Augmentation.** As depicted in Table 3, we execute a series of ablation studies on various tricks implemented during the creation of the dataset. When instructions are not conditioned upon the image content, the SR of the fine-tuned model on the validation set is a mere 36.9%. In contrast, when instructions are generated with conditioning on the image content, the SR on the validation set is enhanced substantially to 81.6%. This uptick can be primarily attributed to the elevated diversity and intricacy of the generated instructions. Moreover, an augmentation of the SR to 91.6% is observed upon introducing context samples into the instructions. This finding underscores the fact that partitioning the chain of actions and allocating them to the instruction and response can strengthen the model's comprehension of the tool. It is noteworthy to

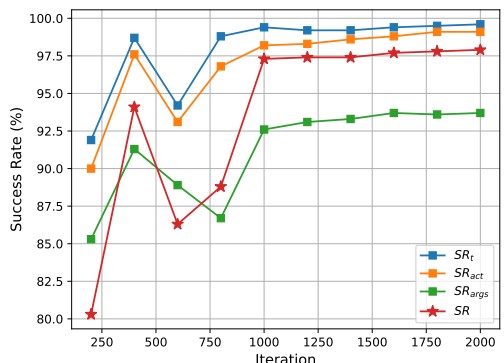

Figure 4: Performance variation curve with the fine-tuning iteration.

mention that with the incorporation of negative samples into the generated instructions, the SR increases to 94.1%. This outcome can be traced back to the propensity of the model, when trained exclusively with positive samples, to bias toward tool invocation. This tendency consequently diminishes the capacity to discern the appropriate cases for tool usage. Adding negative samples equips the model with the ability to determine when to use tools.

**Model Scales.** We attempt experiments with models at different scales. The results in Table 4 demonstrate that after fine-tuned on the generated dataset, Vicuna-7B [12] is also capable of invoking tools in a fixed format. Specifically, under a zero-shot setting, Vicuna-7B achieves only a 4.5% SR. By contrast, after fine-tuning, it can achieve an SR of 92.9%.

**Tuning Iteration.** We increase the number of iterations for fine-tuning and present the results in Figure 4. Notably, during the range of iterations from 400 to 800, the model's performance demonstrates substantial fluctuations in tool invocation. However, subsequent to this range, there is a steady improvement in $SRt$, $SRact$, $SR_{args}$, and SR. This indicates that the model progressively adapts to the dataset, enhancing its capability to invoke tools.

## 4.4 Case Study

Figure 5 presents a comparative analysis of our model with Visual ChatGPT [5] and LLaVa [37]. When an image is submitted by the user alongside the instruction "*Generate a picture of real people based on the edge*", Visual ChatGPT delivers an image that exhibits a weak correlation with the given instruction. Owing to its inability to generate images, LLaVa only returns a caption. In contrast, our model produces an accurate result, thereby evidencing that the tool-related instruction tuning method proposed in this paper can effectively instruct language models in the correct usage of tools. In Figure 6, we further demonstrate that the Vicuna-13B fine-tuned on GPT4Tools is capable of finishing some visual commands by invoking visual tools. This finding indicates that imparting knowledge to

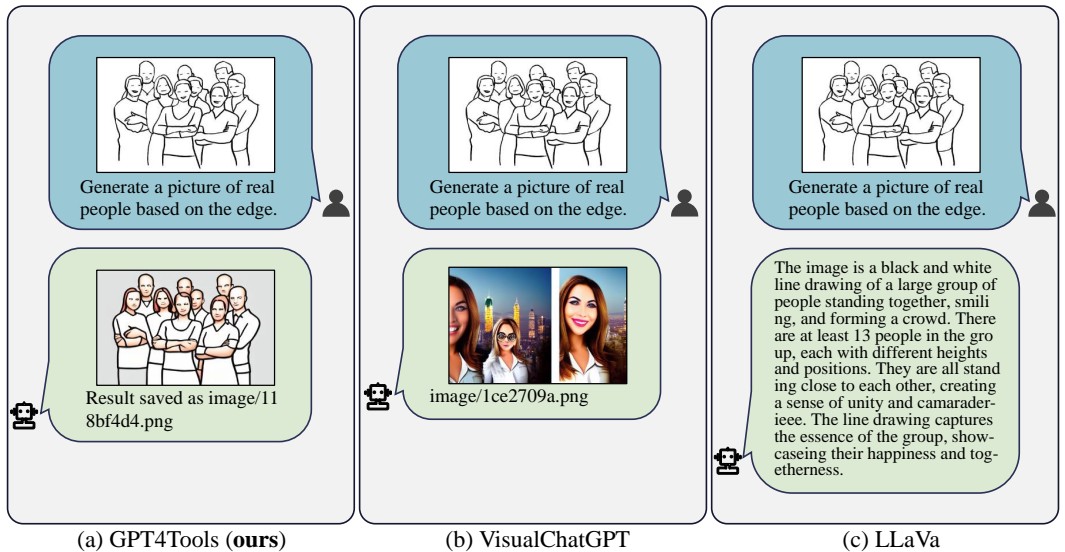

Figure 5: Comparison with other models. Our GPT4Tools responds correctly, while Visual Chat-GPT [5] replies with the wrong image, and LLaVa [37] can not generate the image.

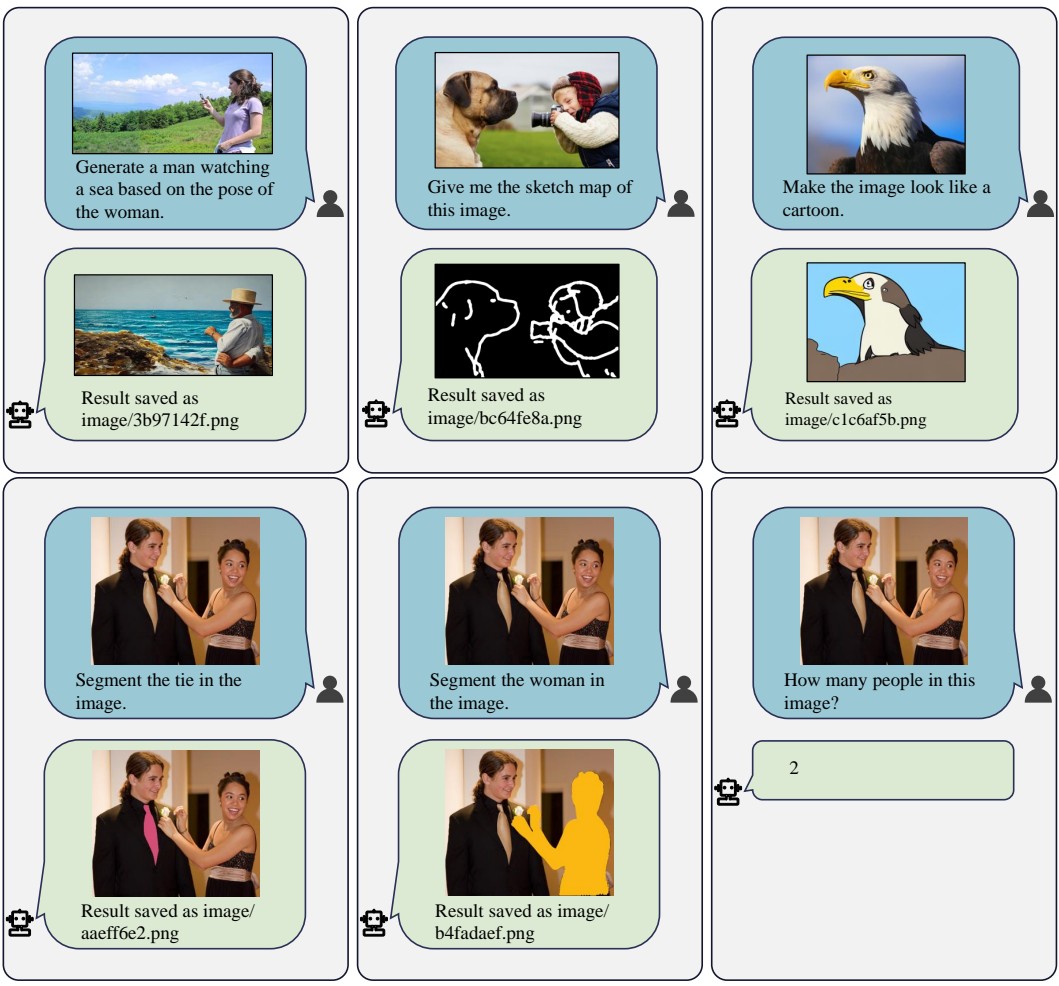

Figure 6: Cases of invoking tools from Vicuna-13B [12] fine-tuned on our GPT4Tools.

Table 5: Results of GPT4Tools using Top-K related tools. The total number of tools equals 23.

| Model | Retrieval Top-K | $SR_t$ | $SR_{act}$ | $SR_{args}$ | SR |
|---|---|---|---|---|---|
| Vicuna-13B [12] | – | 69.2 | 25.1 | 25.2 | 12.4 |
| w/ GPT4Tools | 1 | 87.0 | 55.8 | 57.5 | 54.0 |
| w/ GPT4Tools | 2 | 93.1 | 70 | 69.5 | 67.8 |
| w/ GPT4Tools | 3 | 95.8 | 74.4 | 72.9 | 73.1 |
| w/ GPT4Tools | 23 | 98.7 | 97.6 | 91.4 | 94.1 |

language models regarding the tool invocation could potentially be a way toward the development of a generalist model. More case studies are presented in the Appendix.

## 5 Discussion

Although the proposed GPT4Tools can teach plug-and-play language models to use tools effectively, it still has some limitations. As shown in Figure 2 the success rate of all models is not $100\%$. Thus, further improvements are still necessary for practical applications. Additionally, GPT4Tools teaches the model to explicitly invoke tools using a verbose and fixed prompt. This approach decreases the computational efficiency since attention-based architectures compute the relationships between all tokens. Besides, with the increased number of tools, the prompt length might surpass the limited context length of LLMs. In this case, we can alternatively utilize a tool retrieval technique to filter out a small set of tools and then apply GPT4Tools-based LLMs for tool selection and invocation. We employ BM25, based on the user input, to retrieve the top-K tools from the defined 23 tools. As shown in Table 5, SR is only $54\%$ while retrieving the top-1 tool. When the number of retrieved tools increases to 3, SR is boosted to $73.1\%$. Although the retrieval strategy can mitigate the reliance on long context models for a large number of tools to some extent, its SR can not match the original model. This result can be attributed to the inabilities of the retriever. Therefore, in the future, it is imperative to build a specialized retriever for the tool name retrieval. Moreover, it should be explored how to enable the model to implicitly invoke various tools instead of using the complex prompt. Nevertheless, our GPT4Tools method provides a viable approach for equipping language models with the ability to use multi-modal tools.

## 6 Conclusion

In this paper, we introduce GPT4Tools, a novel method that enables open-source language models to utilize multi-modal tools efficiently. Specifically, We construct a tool-related instructional dataset by prompting advanced GPT-3.5 conditional on image context. Then, we augment the generated data by introducing negative and context samples. Based on the built dataset, we employ LoRA fine-tuning technology to enhance the tool-usage capability of language models, thus allowing them to handle various visual tasks, e.g., visual comprehension and image generation. Moreover, we propose a benchmark to assess tool usage accuracy from the decision when to use tools, which tools to use, and arguments of invoked tools. In this benchmark, language models tuned with our GPT4Tools perform comparably to GPT-3.5 on unseen tools. We desire the GPT4Tools to pave the way for equipping open-source language models with the ability to use multi-modal tools.

## Acknowledgments

This research is partly supported by the National Key R&D Program of China (Grants No. 2020AAA0108302 & 2020AAA0108303), and Shenzhen Science and Technology Project (Grant No. JCYJ20200109143041798) & Shenzhen Stable Supporting Program (WDZC20200820200655001) & Shenzhen Key Laboratory of next-generation interactive media innovative technology (Grant No. ZDSYS20210623092001004).

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
