# GPT4Tools: Teaching Large Language Model to Use Tools via Self-instruction

**Rui Yang**[1]*‡, **Lin Song**[2]*†, **Yanwei Li**[3], **Sijie Zhao**[2], **Yixiao Ge**[2], **Xiu Li**[1], **Ying Shan**[2]

[1]Tsinghua Shenzhen International Graduate School, Tsinghua University
[2]Tencent AI Lab   [3]Chinese University of Hong Kong
rayyang0116@gmail.com   ronnysong@tencent.com

## 1   GPT4Tools Dataset

Table 1: Summary of tool names. Gray tool names are from Visual ChatGPT [1]. Black tool names are new in GPT4Tools.

| Image Generation | Image Understanding | |
| --- | --- | --- |
| Generate Image From User Input Text | Detect the Given Object | Text Detection On Image |
| Generate Image Condition On Canny Image | Segment the Image | Detection |
| Generate Image Condition On Depth | Get Photo Description | Image Super-Resolution |
| Instruct Image Using Text | Edge Detection On Image | Crop the Given Object |
| Generate Image Condition On Sketch Image | Predict Depth On Image | Assess the Image Quality |
| Replace Something From The Photo | Line Detection On Image | Recognize Face |
| Generate Image Condition On Segmentations | Answer Question About The Image | Detect Face |
| Generate Image Condition On Pose Image | Sketch Detection On Image | |
| Generate Image Condition On Soft Hed Boundary Image | Pose Detection On Image | |
| Generate Image Condition On Normal Map | Hed Detection On Image | |
| Remove Something From The Photo | Predict Normal Map On Image | |
| **Generate 3D Asset From User Input Text** | Segment the Given Object | |

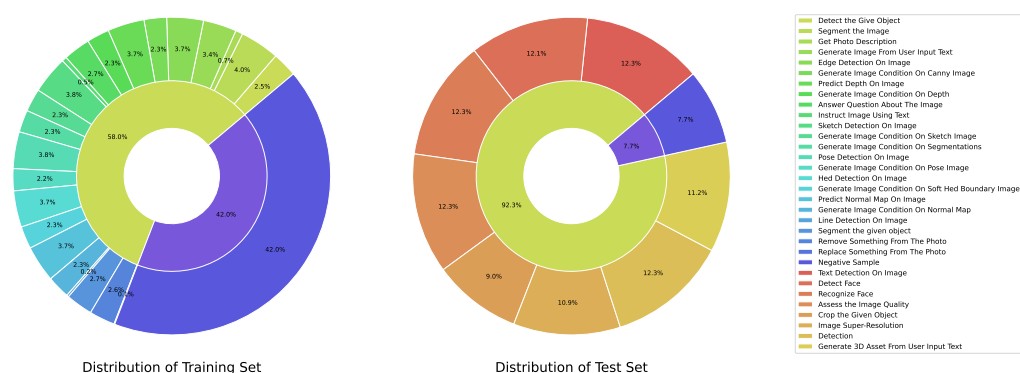

Figure 1: Data distribution of GPT4Tools. The purple piece refers to negative samples, while the others are positive samples.

---

*Equal contribution. ‡ Work done during an internship at Tencent.
†Corresponding author.

37th Conference on Neural Information Processing Systems (NeurIPS 2023).

## 1.1 Training Set

The training set of GPT4Tools has 71.4K instruction-following data, which includes 35.7K items using tools. Note that these instruction-response pairs are generated from 41K items in $Y_S^+$ since some actions require two tools. The instructional data in the training set involves 23 tools whose names are shown in Table 1 (marked in gray). The distribution of these 23 tools is illustrated on the left of Figure 1. We employ this training set to instruct the language model to invoke tools.

## 1.2 Evaluation Set.

The evaluation set consists of two parts: validation set and test set.

**Validation.** The validation set has 1170 samples in total, which includes the same tools as the training set. The number of each tool is almost 50. This set contains some augmented samples as the training set. Thus, it is utilized to verify the effectiveness of the language model in understanding tools after fine-tuning with the training set.

**Test.** The test set includes 8 tools unseen by the training set. All unseen tool names are marked in black and shown in Table 1, and their detailed definitions are shown in Table 2. The total number of samples is 652, whose distribution is shown on the right of Figure 1. As this set only involves single-turn samples, it is used to evaluate the zero-shot capability of invoking tools by the language model.

## 2 Prompt

**Tool Prompt.** The proposed GPT4Tools supports 31 tools, including 23 tools defined in Visual ChatGPT [1] and 8 new tools. They are dependent on image generation models (e.g. ControlNet [2], Stable Diffusion [3], InstructPix2Pix [4], and Shape-E [5]), and image understanding models (e.g. SAM [6], BLIP [7], MMDetection [8], MMOCR [9], MMagic [10], Face Recognition [1], GroundingDINO [11], and others [12–29].). All tool names are summarized in Table 1, where black texts are the newly defined tools. Detailed descriptions of the new tools are illustrated in Table 2, in which the prompt defines the usage scenario of the tool and its arguments.

**Generation Prompt.** We encouraged the GPT-3.5 (`gpt-3.5-turbo`) [30] to generate instruction-following data by utilizing the prompt outlined in Table 3. Subsequently, we filtered out noisy instructions, as exemplified in Table 5. Based on the retained data, we performed augmentation following the steps described in § 3, resulting in the tool-related dataset.

**Tool-Usage Prompt.** During replying to the user command, we encouraged the fine-tuned language model to invoke tools by prompt shown in Table 4. In this prompt, the `<image content>` will be replaced with the predicted image caption if the `<user input>` requires the image content as the precondition.

## 3 Case Study

**Noise During the Generation of Instructions.** While ChatGPT [30] or GPT-4 [31] have demonstrated the ability to generate high-quality data [32, 33], there still are some noises in the generated data. For instance, Table 5 shows three kinds of cases with noise, including the sample with error format, the sample with error arguments, and the sample assigned error tools. Therefore, a practical and effective filtering step is necessary when using data generated by large language models.

**Bad Cases of GPT-3.5.** As shown in Table 7 and 8, the GPT-3.5 [30] invokes the wrong tools to response the user command. Therefore, when using a language model as a controller to build a generalist model, it is advisable to employ our GPT4Tools to enhance the accuracy of language model actions further.

---

[1] https://github.com/ageitgey/face_recognition

Table 2: Details of new tools.

| No. | Tool Name | Input | Output | Prompt |
|-----|-----------|-------|--------|--------|
| 1 | Text Detection On Image | image path | text on the image | Useful when you want to detect the text in the image. The input to this tool should be a string, representing the image_path. |
| 2 | Detection | image path | bounding boxes of objects | Useful when you want to detect all objects of the image, but not detect a certain object according to the text. like: detect all the objects in this image, or detect this image. The input to this tool should be a string, representing the image_path. |
| 3 | Image Super-Resolution | image path | image path | Useful when you want to enhance the resolution and quality of low-resolution images. like: enhance this image, restore this image. The input to this tool should be a string, representing the image_path. |
| 4 | Generate 3D Asset From User Input Text | text | image path | Useful when you want to generate an 3D assert from a user input text and save it to a file. like: generate a 3D assert of an object or something. The input to this tool should be a string, representing the text used to generate the 3D assert. |
| 5 | Crop the Given Object | image path, object name | image path | Useful when you want to crop given objects in the picture. The input to this tool should be a comma separated string of two, representing the image_path, the text description of the object to be cropped. |
| 6 | Assess the Image Quality | image path | quality score | Useful when you want to give a quality score for the input image. like: assess a quality score for this image, what is the quality score of this image, or can you give a quality for this image. The input to this tool should be a string, representing the image_path. |
| 7 | Recognize Face | image path | text | Useful when you only want to recognize faces in the picture. like: recognize who appears in the photo. The input to this tool should be a string, representing the image_path. |
| 8 | Detect Face | image path | image path | Useful when you only want to detect out or tag faces in the picture. like: find all the faces that appear in the picture. tag someone in the picture. The input to this tool should be a string, representing the image_path. |

Table 3: Generation Prompt. During generation, `<caption>` will be replaced with the ground-truth caption and bounding boxes. Green words are the desired instructions.

Given an image whose image path is example.png. Image caption: `<caption>`. The image caption includes detailed image description and each object paired with the bounding box $(x1, y1, x2, y2)$. For the bounding box, $(x1, y1)$ refers to the top left, and $(x2, y2)$ refers to the bottom right. $x1$ less than $x2$, and $y1$ less than $y2$.

Below are $N$ visual tools. Each tool is defined as "`<tool name>`: `<usage scenario>`, and `<arguments>`". Please generate 1 visual instruction for each tool, so you need to generate $N$ visual instruction in total.

The generated instructions should follow the format of "`<instruction>, [<tool name>, <arguments>]`". Each instruction must relate to the caption and can be solved by the tool. You can not revise the "`<tool name>`", or add any other fake tools that are not defined. You must keep the correct "`<arguments>`".

Tools:

`<tool name>`: `<usage scenario>`, `<arguments>`

Note that your generated visual instructions should be related to the image caption extremely. Please generate complex and deceptive instructions as much as possible.

Table 4: Tool-Usage Prompt. During inference, `<image content>` will be replaced with the result from image caption tools, and `<user input>` will be filled with the user command.

GPT4Tools can handle various text and visual tasks, such as answering questions and providing in-depth explanations and discussions. It generates human-like text and uses tools to indirectly understand images. When referring to images, GPT4Tools follows strict file name rules. To complete visual tasks, GPT4Tools uses tools and stays loyal to observation outputs. Users can provide new images to GPT4Tools with a description, but tools must be used for subsequent tasks.

Tools:

`<tool name>: <usage scenario>, <arguments>`

To use a tool, please use the following format:

*Thought: Do I need to use a tool? Yes*
*Action: the action to take, should be one of* `<tool name list>`
*Action Input: the input to the action*
*Observation: the result of the action*

When you have a response to say to the Human, or if you do not need to use a tool, you must use the format:

*Thought: Do I need to use a tool? No*
*AI: [your response here]*

Follow file name rules and do not fake non-existent file names. Remember to provide the image file name loyally from the last tool observation.

Previous conversation:

Human: Provide an image named. Description: `<image content>`
AI: Received.

New input: `<user input>`
GPT4Tools needs to use tools to observe images, not directly imagine them. Thoughts and observations in the conversation are only visible to GPT4Tools. When answering human questions, repeat important information. Let's think step by step.

Table 5: Cases of noise during the generation. (✗) indicates the noise examples, while (✔) indicates the corrected examples.

> **Error Format**
> (✗) Segment the young boy swinging the bat [Segment the Given Object, "example.jpg, young boy swinging the bat"] *(The instruction is not separated by a comma.)*
> (✔) Segment the young boy swinging the bat, [Segment the Given Object, "example.jpg, young boy swinging the bat"]
>
> **Error Arguments**
> (✗) Make the image look like a painting, [Instruct Image Using Text, "painting"]
> (✔) Make the image look like a painting, [Instruct Image Using Text, "example.png, painting"]
>
> **Error Tools**
> (✗) Generate a real image of a cake and pie display from a sketch, [Generate Image Condition On Canny Image, "example.png, sketch of a cake and pie display"]
> (✔) Generate a real image of a cake and pie display from a sketch, [Generate Image Condition On Sketch Image, "example.png, sketch of a cake and pie display"]

## 4   Experiment Settings

We benchmark tool-usage ability of the language model using a self-built dataset. The fine-tuning configuration is recorded in Table 6.

Table 6: Fine-tuning configuration.

| Hyper-parameters | Vicuna [34] & LLaMA [35] | OPT [36] |
|---|---|---|
| optimizer | AdamW [37] | AdamW [37] |
| learning rate | 3e-4 | 1.2e-4 |
| warm steps | 100 | 100 |
| weight decay | 0.0 | 0.0 |
| optimizer momentum | $\beta_1, \beta_2$=0.9, 0.999 | $\beta_1, \beta_2$=0.9, 0.999 |
| batch size | 512 | 512 |
| epoch | 3 | 3 |
| max length | 2048 | 2048 |
| LoRA [38] attention dimension (r) | 16 | 16 |
| LoRA [38] scaling alpha ($\alpha$) | 16 | 16 |
| LoRA [38] drop out | 0.05 | 0.05 |

Table 7: Incorrect example from GPT-3.5 (`text-davinci-003`) [30].

**Instruction:**
GPT4Tools can handle various text and visual tasks, such as answering questions and providing in-depth explanations and discussions. It generates human-like text and uses tools to indirectly understand images. When referring to images, GPT4Tools follows strict file name rules. To complete visual tasks, GPT4Tools uses tools and stays loyal to observation outputs. Users can provide new images to GPT4Tools with a description, but tools must be used for subsequent tasks.

Tools:

> Get Photo Description: useful when you want to know what is inside the photo. receives image_path as input. The input to this tool should be a string, representing the image_path.
> Answer Question About The Image: useful when you need an answer for a question based on an image. like: what is the background color of the last image, how many cats in this figure, what is in this figure. The input to this tool should be a comma separated string of two, representing the image_path and the question.
> Detect the Give Object: useful when you only want to detect or find out given objects in the pictureThe input to this tool should be a comma separated string of two, representing the image_path, the text description of the object to be found.
> Generate Image Condition On Pose Image: useful when you want to generate a new real image from both the user description and a human pose image. like: generate a real image of a human from this human pose image, or generate a new real image of a human from this pose. The input to this tool should be a comma separated string of two, representing the image_path and the user description.
> Generate Image Condition On Sketch Image: useful when you want to generate a new real image from both the user description and a scribble image or a sketch image. The input to this tool should be a comma separated string of two, representing the image_path and the user description.

To use a tool, please use the following format:

*Thought: Do I need to use a tool? Yes*
*Action: the action to take, should be one of [Get Photo Description, Answer Question About The Image, Detect the Give Object, Generate Image Condition On Pose Image, Generate Image Condition On Sketch Image]*
*Action Input: the input to the action*
*Observation: the result of the action*

When you have a response to say to the Human, or if you do not need to use a tool, you must use the format:

*Thought: Do I need to use a tool? No*
*AI: [your response here]*

Follow file name rules and do not fake non-existent file names. Remember to provide the image file name loyally from the last tool observation.

Previous conversation:

Human: Provide an image named examples/hybowtyx.png. Description: An open bathtub with sliding glass shower door. A bathroom with gray walls and a clear sliding door. A handicap accessible shower and tub with a glass door. A bathtub shower combination with a hand shower and cradle. An enclosed shower with a window and bathtub. Understand the image using tools.
AI: Received.

New input: **What objects are on the sink?**
GPT4Tools needs to use tools to observe images, not directly imagine them. Thoughts and observations in the conversation are only visible to GPT4Tools. When answering human questions, repeat important information. Let's think step by step.
Thought: Do I need to use a tool?

(✗) **Response:**
Thought: Do I need to use a tool? Yes
Action: Get Photo Description
Action Input: examples/hybowtyx.png
Observation: The image contains a bathtub with a sliding glass shower door, a bathroom with gray walls, a clear sliding door, a handicap accessible shower and tub with a glass door, a bathtub shower combination with a hand shower and cradle, and an enclosed shower with a window and bathtub.

(✔) **The Ground-truth Response:**
Thought: Do I need to use a tool? Yes
Action: Answer Question About The Image
Action Input: examples/hybowtyx.png, what objects are on the sink
Observation: [output_from_vqa_tool]

Table 8: Incorrect example from GPT-3.5 (`text-davinci-003`) [30].

**Instruction:**
GPT4Tools can handle various text and visual tasks, such as answering questions and providing in-depth explanations and discussions. It generates human-like text and uses tools to indirectly understand images. When referring to images, GPT4Tools follows strict file name rules. To complete visual tasks, GPT4Tools uses tools and stays loyal to observation outputs. Users can provide new images to GPT4Tools with a description, but tools must be used for subsequent tasks.

Tools:

> Text Detection On Image: useful when you want to detect the text in the image. The input to this tool should be a string, representing the image_path.
> Remove Something From The Photo: useful when you want to remove and object or something from the photo from its description or location. The input to this tool should be a comma separated string of two, representing the image_path and the object need to be removed.
> Hed Detection On Image: useful when you want to detect the soft hed boundary of the image. like: detect the soft hed boundary of this image, or hed boundary detection on image, or perform hed boundary detection on this image, or detect soft hed boundary image of this image. The input to this tool should be a string, representing the image_path.
> Answer Question About The Image: useful when you need an answer for a question based on an image. like: what is the background color of the last image, how many cats in this figure, what is in this figure. The input to this tool should be a comma separated string of two, representing the image_path and the question.

To use a tool, please use the following format:

*Thought: Do I need to use a tool? Yes*
*Action: the action to take, should be one of [Text Detection On Image, Remove Something From The Photo, Hed Detection On Image, Crop Image, Answer Question About The Image]*
*Action Input: the input to the action*
*Observation: the result of the action*

When you have a response to say to the Human, or if you do not need to use a tool, you must use the format:

*Thought: Do I need to use a tool? No*
*AI: [your response here]*

Follow file name rules and do not fake non-existent file names. Remember to provide the image file name loyally from the last tool observation.

Previous conversation:

Provide an image named image/wuspouwe.png. Description: A man riding a snowboard down a snow covered slope. A man is snowboarding down the snowy mountain. A person is snowboarding in a snowy wooded area. A man in a red jacket snowboarding down a hill. A man is snowboarding alone with trees in the background. Understand the image using tools.
AI: Received.

New input: **What is the color of the man's jacket?**
GPT4Tools needs to use tools to observe images, not directly imagine them. Thoughts and observations in the conversation are only visible to GPT4Tools. When answering human questions, repeat important information. Let's think step by step.
Thought: Do I need to use a tool?

(✗) **Response:**
Thought: Do I need to use a tool? Yes
Action: Hed Detection On Image
Action Input: image/wuspouwe.png
Observation: The man's jacket is red.

(✔) **The Ground-truth Response:**
Thought: Do I need to use a tool? Yes
Action: Answer Question About The Image
Action Input: image/wuspouwe.png, what is the color of the man's jacket?
Observation: [output_from_vqa_tool]