# OpenReview forum: "GPT4Tools: Teaching Large Language Model to Use Tools via Self-instruction"
_NeurIPS.cc/2023/Conference — NeurIPS 2023 poster_

### Official Review · Reviewer_F8mq · 2023-07-05

**Soundness:** 3 good
**Presentation:** 3 good
**Contribution:** 2 fair
**Rating:** 5
**Confidence:** 4

**Summary:**

The goal of this paper is to utilize locally available models, such as vicuna and opt, to learn the tool utilization capability of black-box models like GPT-3.5. They use the large models to generate instructions and samples using image context and apply instruction tuning on these data with the fine-tunable models. Various data augmentation techniques such as negative samples and context samples were also applied. Furthermore, they propose several automated evaluation methods, including the success rate based on the combination of thought, action, and arguments. Experimental results demonstrate significant improvement in tool invocation effectiveness for their fine-tuned models compared to the original backbone models.

**Strengths:**

1. This paper proposes a method for generating instructions using image context, which enables the large language model to consider visual information when generating multimodal instructions.
2. The method of generating negative data has not been considered in previous works.
3. This paper introduces a novel automated evaluation metric that combines multiple success rates.

**Weaknesses:**

1. The method proposed in this paper, which utilizes self-instruction techniques of large models to enable tool using capabilities for primitive models, is not novel. Similar techniques have been proposed in several works, such as tool-llama (https://arxiv.org/abs/2304.08354), tool-alpaca (http://arxiv.org/abs/2306.05301), and gorilla (http://arxiv.org/abs/2305.15334). The authors have not made significant contributions to the SFT method.

2. One of the main contributions of this paper, image context, is not thoroughly elaborated in this paper. The self-instruction method seems to be also similar to that of MM-react and LLaVA.

3. For the evaluation metrics, the success rate of thought is not clearly explained. Thought should be a natural languge text with certain diversity, but it is unclear whether the generated thought needs to be identical to the ground truth thought to achieve success rate 1. If they need to be identical, I doubt that the success rate of thought would not be as high as shown in the experimental results, as the model may generate thought that is semantically consistent but not textually identical to the groundtruth. If they do not need to be identical, it would be helpful to provide details on how the success rate of thought is calculated.

4. Some recent strong multimodal baselines, such as HuggingGPT (http://arxiv.org/abs/2303.17580) and gorilla (http://arxiv.org/abs/2305.15334), are missing in this paper.

**Questions:**

1. Is there a technical error in Equation 1? where it states that "Y" is sampled from the teacher model based on "X" to obtain the prompt. However, based on the textual description, Equation 1 should refer to sampling the instruction based on "X" and "P_t."
2. How the image context was constructed?
3. How is the success rate of thought be computed?

---

> ### Author Rebuttal · Authors · 2023-08-10
>
> **Q1:** *The method proposed in this paper, which utilizes self-instruction techniques of large models to enable tool using capabilities for primitive models, is not novel. Similar techniques have been proposed in several works, such as tool-llama (https://arxiv.org/abs/2304.08354), tool-alpaca (http://arxiv.org/abs/2306.05301), and gorilla (http://arxiv.org/abs/2305.15334). The authors have not made significant contributions to the SFT method.*
> **A1:** We appreciate the comments but believe there exists a big misunderstanding. Our GPT4Tools is the first public work (before all the mentioned works) to enable a primitive language model to use multi-modal tools [51], including demo and codes. Besides, all the mentioned works were pre-printed after the submission deadline. Please note that the mentioned Tool-LLaMa could be [https://arxiv.org/pdf/2307.16789.pdf] instead of the BMTools [https://arxiv.org/abs/2304.08354]. The BMTools relies on the prompt engineering of ChatGPT instead of off-the-shelf language models. By contrast, our GPT4Tools aims to empower open-sourced language models invoking tools through self-instruction tuning.
>
> Our contributions mainly lie in three points:
> (1) Our method first enables primitive open-source language models to use multi-modal tools, eliminating the dependence on advanced proprietary LLMs like ChatGPT.
> (2) We design a new approach based on multi-modal contexts for self-instruction and augmentation, which significantly promote multi-modal tool usage and can be deployed in different approaches.
> (3) We propose a new benchmark to assess the effectiveness of using tools, and our method shows remarkable improvements.
>
> [51] Yin S, Fu C, Zhao S, et al. A Survey on Multimodal Large Language Models[J]. arXiv preprint arXiv:2306.13549, 2023.
>
> **Q2:** *One of the main contributions of this paper, image context, is not thoroughly elaborated in this paper. The self-instruction method seems to be also similar to that of MM-react and LLaVA.*
> **A2:** Please note that our GPT4Tools is the concurrent work with MM-REACT and LLaVa. Besides, as far as we known, the MM-REACT is based on prompt engineering instead of self-instructions, and the LLaVa aims to achieve a better VQA model rather than pursue the capability of invoking tools. The LLaVa can be also a tool in our GPT4Tools to enhance the visual understanding. In addition, the image context is consist of the image captions and the coordinates of objects. As presented in Section 3.1 and Table 10 of supplementary material, we use these image contexts to build prompt for ChatGPT and then generate tool-related user instructions. We will clarify it in the revision.
>
> **Q3:** *For the evaluation metrics, the success rate of thought is not clearly explained. Thought should be a natural languge text with certain diversity, but it is unclear whether the generated thought needs to be identical to the ground truth thought to achieve success rate 1. If they need to be identical, I doubt that the success rate of thought would not be as high as shown in the experimental results, as the model may generate thought that is semantically consistent but not textually identical to the groundtruth. If they do not need to be identical, it would be helpful to provide details on how the success rate of thought is calculated.*
> **A3:** Different from many open-ended tasks, invoking tools need to strictly follow a pre-defined interaction format. The model arbitrarily outputting in unexpected formats, such as wrong tool names and number of parameters, will lead to failure in properly invoking the tools. As presented in Section 3.3, we evaluate the models from three aspects: whether a tool is needed, tool type, and tool parameters. For open-ended text in the tool parameters, we adopt BLEU metric with threshold to assess correctness.
>
> **Q4:** *Some recent strong multimodal baselines, such as HuggingGPT (http://arxiv.org/abs/2303.17580) and gorilla (http://arxiv.org/abs/2305.15334), are missing in this paper.*
> **A4:** Please refer to Q1. Besides, we evaluate our model with the Visual-ChatGPT in Table 2 and Table 5, which uses the same GPT-3.5 LLM with the HuggingGPT.
>
> **Q5:** *Is there a technical error in Equation 1? where it states that "Y" is sampled from the teacher model based on "X" to obtain the prompt. However, based on the textual description, Equation 1 should refer to sampling the instruction based on "X" and "P_t."*
> **A5**: As shown in Table 10 of supplementary material, the P_t is the system prefix prompt, which is made up of the basic task definition, the usage description of tools and the placeholder for image context. We utilize ChatGPT to generate diverse user instructions by placing different image contexts into the placeholder of the system prompt. More details will be added into the revision.
>
> **Q6:** *How the image context was constructed?*
> **A6:** Please refer to Q2. As illustrated in Figure 1 and Table 10 of supplementary material, the image context is consist of the image captions and the coordinates of objects. We use these image contexts to build prompt for ChatGPT and then generate tool-related user instructions. We will clarify it in the revision.
>
> **Q7:** *How is the success rate of thought be computed?*
> **A7:** Please refer to Q3. As presented in Section 3.3, we evaluate the models from three aspects: whether a tool is needed, tool type, and tool parameters. For open-ended text in the tool parameters, we adopt BLEU metric with threshold to assess correctness.

---

> > ### Comment · Reviewer_F8mq · 2023-08-20
> > **Final review on Submission1848 by Reviewer F8mq**
> >
> > Thank you to the authors for their rebuttal. The authors has explained its difference in using open-source language models to use multi-modal tools and the first work to conduct this. It appears that the authors have addressed the weakness in their response, so I revise my score from 4 to 5. Just as mentioned in Reviewer MACc, I also remain the score at this level duo to that the limited novelty of the proposed method.

---

### Official Review · Reviewer_MACc · 2023-07-06

**Soundness:** 2 fair
**Presentation:** 2 fair
**Contribution:** 4 excellent
**Rating:** 5
**Confidence:** 4

**Summary:**

This work propose GPT4Tools, a self-instruction approach to teach large language models to use tools to solve vision tasks. The authors construct an instruction-following dataset by ChatGPT from the combinations of a set of images and 23 image-related tools. In addition, they augment the dataset to enable to capability to not using tools or using tools in multiple turns. Then they apply Low-Rank Adaptation (LoRA) to open-source LLMs like LLaMa and Vicuna to enable them to use tools. They also propose to evaluate the model's performance on tool-using by measureing the success rate from multiple ascepts. The empirical results show that GPT4Tools can successfully utilize the tools and also have capability to use new tools.
The study introduces GPT4Tools, a self-instruction method that teaches large language models (LLMs) to utilize tools for solving vision tasks. The authors establish an instruction-following dataset generated by ChatGPT using combinations of images and 23 image-related tools. The dataset is augmented to allow for the capability of non-usage of tools or multi-turn tool usage. The researchers then apply Low-Rank Adaptation (LoRA) to open-source LLMs, like LLaMa and Vicuna, enabling these models to use tools. The paper also proposes an evaluation measure for the model's success rate from multiple aspects. Empirical results reveal that GPT4Tools can effectively use tools and demonstrate the capability to adopt new tools.

**Strengths:**

- The paper presents an innovative approach to teaching LLMs to use tools for visual tasks through self-instruction
- The impact of enabling LLMs to utilize tools could be substantial and pervasive.

**Weaknesses:**

- The paper falls short in providing a comparison with VisProg[1], which also instructs LLMs to use visual tools through few-shot demonstrations. Although GPT4Tools has the benefit of not utilizing proprietary LLMs in inference, a performance comparison would enrich the paper.
- The evaluation method appears less compelling because both the training and testing data are generated by using the specific set of tools. Evaluation on standard benchmarks, such as GQA as in [1], could be more convincing.
- The paper could benefit from enhanced clarity in its descriptions and explanations. Suggestions include:
    - Clarifying the variables P_t, X_C, Y, possibly through their corresponding parts in Figure 1 or mentioning corresponding examples in the appendix.
    - The subscript of P_t could be confusing as the total number of tools is defined as N.
    - The abundant font colors in Figure 1 and Figure 3 may confuse readers, and it would be helpful to indicate which parts are generated by LLMs
- (Minor) Some crucial details appear to be omitted, such as the process of extracting X_C from images, the image source, and the filtering process of the dataset.
- (Minor) Typos:
    - Page 6, line 172: "and and"
    - Page 6, line 173: ""items.."

[1] Gupta, Tanmay, and Aniruddha Kembhavi. "Visual programming: Compositional visual reasoning without training." Proceedings of the IEEE/CVF Conference on Computer Vision and Pattern Recognition. 2023.

**Questions:**

- Can GPT4Tools effectively do multi-turn planning, given the multi-turn examples in the augmented data? Why does the evaluation data only contain single-turn examples?
- How scalable is GPT4Tools when using a larger number of tools?
- (Minor) What the numerical codes (e.g., 179.44, 105.55) represent in the image content shown in Figure 1?

**Limitations:**

While the authors discuss potential overfitting for longer tuning iterations and the inferior performance of OPT-13B, more detailed discussions regarding the limitations of the proposed evaluation method, the multi-turn capability, and the scalability of GPT4Tools would add value.

---

> ### Author Rebuttal · Authors · 2023-08-10
>
> **Q1:** *How effectively can the proposed instruction data collection procedure generalize to other domains?* \
> **A1:** Different from VisProg which utilizes few-shot prompt engineering, our method uses a smaller open-source language model and zero-shot manner. As shown in Table 2, our method demonstrates good generalization capability on unseen tools. Furthermore, to further validate efficacy, we conducted ablation studies on GQA following the VisProg by creating a subset for evaluation. As shown in Table 15, the GPT4Tools further improve the accuracy by 3.4% over the VILT-VQA baseline model. We will add more experiments into the revision.
>
> Table 15: Comparison on GQA Dataset.
>  **Method** | **Accuracy** |
> |:-------------------:|:--------:|
>  VITL-VQA                 | 47.8       |
>  GPT4Tools                | 51.2       |
>
>
> **Q2:** *The paper could benefit from enhanced clarity in its descriptions and explanations.*
> **A2:** We appreciate the valuable suggestions! Accordingly, we will improve the presentation by clarifying symbols and font colors of figures in the revision.
>
> **Q3:** *Some crucial details appear to be omitted, such as the process of extracting X_C from images, the image source, and the filtering process of the dataset.*
> **A3**: All images utilized in GPT4Tools are sourced from the training set of COCO [43]. The X_C consists of ground-truth captions and bounding boxes with tags corresponding to the images, which are used to construct prompts for ChatGPT to generate data. We then utilize postprocessing to filter out data with: duplicate instructions, incorrectly formatted instructions, calls with incorrect tool names, and calls with incorrect tool-arguments formats. The comprehensive description about the image source and filtering procedure will be provided in the revision. Meanwhile, the corresponding code will be made available.
>
> **Q4:** *Can GPT4Tools effectively do multi-turn planning, given the multi-turn examples in the augmented data? Why does the evaluation data only contain single-turn examples?*
> **A4:** Yes, GPT4Tools can perform multi-turn planning. For example, as shown in Figure 7 of supplementary material, the user instruction in a sample "Generate a man watching a sea based on the pose of the woman." requires the following multi-turn planning: detect the woman in the image, extract the pose of the woman, generate an image of a man watching the sea using the pose. In the GPT4Tools, we term these multi-turn planning samples as context samples, which are also included in the validation set.
>
> **Q5:** *How scalable is GPT4Tools when using a larger number of tools?*
> **A5:** LLMs are currently evolving towards supporting long context, for example Long-form [49] supports 1B token inputs. This makes the prompts for a massive number of tools not an unresolvable issue in the future. In addition, for models with limited context length, we can use tool retrieval techniques to filter out the a small subset of toolsets that may be involved for the user input, and then apply GPT4Tools for inference. To demonstrate effectiveness, we utilized BM25[50] to retrieve the top K tools from 23 tools based on the user input for inference, whose results are shown in Table 14. This strategy can mitigate the reliance on long context models for a large number of tools to some extent. We will provide more experiments on this in the revision.
>
> Table 14: GPT4Tools using Top-K related tools.
> | Model | SR_t | SR_act | SR_args | SR   |
> |-----------------|------|--------|---------|------|
> | Vicuna-13B   | 69.2 | 24.5   | 24.6    | 12.3 |
> | + GPT4Tools (Top-1)  | 87.0 | 55.8   | 57.5    | 54   |
> | + GPT4Tools (Top-2)  | 93.1 | 70     | 69.5    | 67.8 |
> | + GPT4Tools (Top-3)  | 95.8 | 74.4   | 72.9    | 73.1 |
> | + GPT4Tools (All)  | 99.2 | 98.1   | 92.9    | 89.8 |
>
> [49] Ding J, Ma S, Dong L, et al. LongNet: Scaling Transformers to 1,000,000,000 Tokens[J]. arXiv preprint arXiv:2307.02486, 2023.
> [50] Robertson, S.E., S. Walker, S. Jones, M.M. Beaulieu, M.Gatford. Okapi at TREC-3. TREC-3, (1994), p. 109-126

---

> > ### Comment · Reviewer_MACc · 2023-08-18
> >
> > I appreciate the authors for the comprehensive response and for conducting additional experiments. Many of my initial concerns have been addressed, prompting me to revise my score from 4 to 5. However, the score remains at this level due to the limited novelty of the proposed method (instruction-tuning) and the clarity issues mentioned above. Thank you for the understanding.

---

### Official Review · Reviewer_VYCC · 2023-07-06

**Soundness:** 3 good
**Presentation:** 3 good
**Contribution:** 3 good
**Rating:** 6
**Confidence:** 3

**Summary:**

This paper utilizes ChatGPT to generate instruction data for training moderately-sized open-source models to use tools. The authors fine-tune LLMs using LoRA and demonstrate their effectiveness on various visual problems. Additionally, the paper establishes a benchmark for evaluating the LLMs' tool usage ability.

**Strengths:**

1. Using ChatGPT, this paper represents an early effort to consider visual information to produce diverse instruction data that trains moderately-sized open-source models to use tools, resulting in effective learning outcomes.

2. This paper sets a benchmark standard for evaluating the competency of LLMs in tool usage.

3. The results of the experiment demonstrate the efficacy of instruction tuning, as evidenced by the superior performance of the tuned 13B model in comparison to the 175B GPT-3.5 model. Additionally, the study highlights the framework's capacity for generalization to unseen tools.

**Weaknesses:**

1. It would be intriguing to understand the reasons behind the significant discrepancy in generalization ability between OPT and LLaMA in Table 2.
2. The prefix prompt used in the study includes both system messages and tool definitions. Although the current context length of open LLMs appears to be sufficient for the 23 tools examined in the paper, it may not be enough if a broader range of tools is considered. Therefore, the present framework would need to address this limitation if it is to accommodate a more diverse range of tools.

**Questions:**

please refer to Weaknesses

**Limitations:**

please refer to Weaknesses

---

> ### Author Rebuttal · Authors · 2023-08-10
>
> **Q1:** *It would be intriguing to understand the reasons behind the significant discrepancy in generalization ability between OPT and LLaMA in Table 2.*
> **A1:** Compared to Vicuna and LLaMA, one possible reason is that OPT has weaker capabilities for multi-turn interaction. There are many samples in Validation-II that require multi-turn tool invocations, and OPT demonstrates poor generalization performance on these samples. In contrast, Vicuna, which is optimized through multi-turn dialogues, performs the best. On the contrary, as shown in Table 5 of supplementary material, OPT exhibits good generalization performance on Validation III which only contains single-turn interactions.
>
> **Q2:** *The prefix prompt used in the study includes both system messages and tool definitions. Although the current context length of open LLMs appears to be sufficient for the 23 tools examined in the paper, it may not be enough if a broader range of tools is considered. Therefore, the present framework would need to address this limitation if it is to accommodate a more diverse range of tools.*
> **A2:** LLMs are currently evolving towards supporting long context, for example Long-form [49] supports 1B token inputs. This makes the prompts for a massive number of tools not an unresolvable issue in the future. In addition, for models with limited context length, we can use tool retrieval techniques to filter out the a small subset of toolsets that may be involved for the user input, and then apply GPT4Tools for inference. To demonstrate effectiveness, we utilized BM25[50] to retrieve the top K tools from 23 tools based on the user input for inference, whose results are shown in Table 14. This strategy can mitigate the reliance on long context models for a large number of tools to some extent. We will provide more experiments on this in the revision.
>
> Table 14: GPT4Tools using Top-K related tools.
> | Model | SR_t | SR_act | SR_args | SR   |
> |-----------------|------|--------|---------|------|
> | Vicuna-13B   | 69.2 | 24.5   | 24.6    | 12.3 |
> | + GPT4Tools (Top-1)  | 87.0 | 55.8   | 57.5    | 54   |
> | + GPT4Tools (Top-2)  | 93.1 | 70     | 69.5    | 67.8 |
> | + GPT4Tools (Top-3)  | 95.8 | 74.4   | 72.9    | 73.1 |
> | + GPT4Tools (All)  | 99.2 | 98.1   | 92.9    | 89.8 |
>
> [49] Ding J, Ma S, Dong L, et al. LongNet: Scaling Transformers to 1,000,000,000 Tokens[J]. arXiv preprint arXiv:2307.02486, 2023.
> [50] Robertson, S.E., S. Walker, S. Jones, M.M. Beaulieu, M.Gatford. Okapi at TREC-3. TREC-3, (1994), p. 109-126

---

### Official Review · Reviewer_witA · 2023-07-08

**Soundness:** 3 good
**Presentation:** 2 fair
**Contribution:** 3 good
**Rating:** 6
**Confidence:** 4

**Summary:**

The authors propose GPT4Tools, a new approach to data collection for tool use, along with a novel evaluation method. In this approach, ChatGPT is utilized as a teacher to generate instruction-following data for vision-language tasks related to tools. In addition to the single-turn samples, the authors suggest synthesizing negative samples to instruct the student model on when not to use a tool. They also propose using context samples to guide the student model in utilizing relevant tools based on contextual information. Experimental results indicate that by employing the proposed successful rate(s), open source models like LLaMA-13B and Vicuna-13B achieve significant performance improvements after LoRA fine-tuning. Furthermore, the authors demonstrate a certain level of capability in utilizing unseen tools after the fine-tuning process.

**Strengths:**

* The proposed procedure for instruction data collection is straightforward and easy to follow. The performance improvement of the open-source models is evident. The effectiveness of the proposed negative samples and context samples is clearly demonstrated in facilitating downstream tool usage.
* The proposed evaluation metric is logical and valuable in establishing a standard for future research in this field.
* The conducted ablation study on data augmentation, model size, and tuning iterations provides valuable insights into understanding the behavior of the model in depth.


**Weaknesses:**

* I believe the work focuses on a straightforward procedure for instruction data collection and a standardized evaluation method. Therefore, I don't identify any significant weaknesses, but rather some points for further discussion:
  * The process of creating synthetic data from ChatGPT and using it to fine-tune smaller models can be seen as a distillation process. Previous studies have demonstrated that, for many tasks, distilled student language models can outperform their teacher language models across various tasks [A1]. What are the authors' thoughts on the relationship between instruction tuning and distillation in the context of tool use?
  * Vision tasks often exhibit stronger interdependencies. How effectively can the proposed instruction data collection procedure generalize to other domains, such as manipulating a vast number of APIs for cloud services, etc?
* I did not find a clear distinction between the two sub-figures in Figure 2. Could the authors emphasize the key area that readers should focus on?
* There is a typo on line 155: "single-tune" should be corrected to "single-turn."


[A1] Distilling Step-by-Step! Outperforming Larger Language Models with Less Training Data and Smaller Model Sizes. ACL 2023.


**Questions:**

Please see the weaknesses section above. Overall I enjoy reading the submission. Since the work mostly discusses instruction data collection procedure, it would be great to expand the discussion between the proposed method and other related fields, such as distillation, other instruction tuning works, etc.

**Limitations:**

Limitations are included

---

> ### Author Rebuttal · Authors · 2023-08-10
>
> **Q1:** *The process of creating synthetic data from ChatGPT and using it to fine-tune smaller models can be seen as a distillation process. Previous studies have demonstrated that, for many tasks, distilled student language models can outperform their teacher language models across various tasks. What are the authors' thoughts on the relationship between instruction tuning and distillation in the context of tool use?*
> **A1:** From the perspective of training data, distillation requires input samples to be given and then fed into a network to generate training targets. On the other hand, self-instruction tuning uses a network to generate both input samples and training targets. Therefore, distillation can also be a way to build training targets for self-instruction tuning. However, the most critical challenge in enabling language models to use tools is the lack of diverse input samples, specifically user instructions. Therefore, we leverage the self-instruction technique to utilize advanced language models in generating user instructions based on image content and tool descriptions. In addition, the effectiveness of distillation is limited by the capabilities of the teacher model, which may introduce much noise into the training targets. However, in our GPT4Tools, since we know in advance the toolchains involved in the input samples, we can greatly reduce the noise in the training targets through heuristic rules.
>
> **Q2:** *Vision tasks often exhibit stronger interdependencies. How effectively can the proposed instruction data collection procedure generalize to other domains, such as manipulating a vast number of APIs for cloud services, etc?*
> **A2:** This is a thoughtful question. In order to enable LLMs to use tools, the following points need to be achieved: judge whether a tool is needed, select the appropriate tool, generate suitable parameters, and interact with the tool in the correct format. For GPT4Tools, we mainly enable the LLM to understand the prompts for toolset definitions and output in the correct format, while selecting the right tools and parameters relies on the own capabilities of models. This allows our method to perform well on unseen toolsets, which is shown in Table 2 and Table 5. We will add more additional tools unrelated vision in the revision.
>
> **Q3:** *A clear distinction between the two sub-figures in Figure 2. Could the authors emphasize the key area that readers should focus on?*
> **A3:** Figure 2 illustrates the diversity comparison of the collected instructional dataset with and without the image content. Without image context priors, language models tend to generate objects of visual instructions towards a small subset, which is reflected in t-SNE as sparser clusters. In contrast, using image context notably increases diversity, reflected in the visualization as denser and more widely distributed results. Therefore, the language model tuned by image-conditioned data has a higher success rate compared to models without the image content (81.6% v.s. 36.9%).

---

> > ### Comment · Reviewer_witA · 2023-08-13
> >
> > Thank you to the authors for their rebuttal. I had similar concerns to those raised by reviewers F8mq and MACc. It appears that the authors have addressed these concerns in their response.  The authors also mostly addressed my concerns as well.

---

### Decision · Program_Chairs · 2023-09-21

**Decision:**

Accept (poster)

**Comment:**

GPT4Tools is proposed: ChatGPT is utilized as a teacher to generate instruction-following data for tool use. In addition to the single-turn samples, negative samples are synthesized to instruct the student model on when not to use a tool. Context samples are used to guide the student model in utilizing relevant tools based on contextual information.

All reviewers agree to accept the paper, though some shared concerns include limited novelty.